# GARLIC: GAussian Representation LearnIng for spaCe partitioning

## Abstract

We present **GARLIC**, a representation learning approach for Euclidean approximate nearest neighbor (ANN) search in high dimensions. Existing partitions tend to rely on isotropic cells, fixed global resolution, or balanced constraints, which fragment dense regions and merge unrelated points in sparse ones, thereby increasing the candidate count when probing only a few cells. Our method instead partitions $\mathbb{R}^d$ into anisotropic Gaussian cells whose shapes align with local geometry and sizes adapt to data density. Information-theoretic objectives balance coverage, overlap, and geometric alignment, while split/clone refinement introduces Gaussians only where needed. At query time, Mahalanobis distance selects relevant cells and localized quantization prunes candidates. This yields partitions that reduce cross-cell neighbor splits and candidate counts under small probe budgets, while remaining robust even when trained on only a small fraction of the dataset. Overall, GARLIC introduces a geometry-aware space-partitioning paradigm that combines information-theoretic objectives with adaptive density refinement, offering competitive recall–efficiency trade-offs for Euclidean ANN search.

## 1 Introduction

Let $X = \{x_i\}_{i=1}^n \subset \mathbb{R}^d$ be a finite point set, $q \in \mathbb{R}^d$ a query, and $\delta_{\mathcal{E}} : \mathbb{R}^d \times \mathbb{R}^d \to \mathbb{R}_{\geq 0}$ the Euclidean distance. For an integer $k \geq 1$, the exact $k$-Nearest Neighbor Search (NNS) problem returns the $k$ closest points $N_k(q) \subseteq X$. Its approximate variant, $k$-ANN, relaxes this by requiring $A_k(q) \subseteq X$, $|A_k(q)| = k$, such that $\max_{a \in A_k(q)} \delta_{\mathcal{E}}(q, a) \leq c\, \delta_{\mathcal{E}}(q, x_{(k)}(q))$, for some approximation factor $c \geq 1$, where $x_{(k)}(q)$ denotes the $k$-th true neighbor of $q$. We restrict attention to Euclidean spaces of hundreds of dimensions, and to indices defined by partitions of $\mathbb{R}^d$ into cells, where a query inspects only a few cells and re-ranks the resulting candidates. Nearest neighbor search in this setting is a canonical problem of high-dimensional geometry and algorithms, with consequences across information retrieval, computer vision, robotics, and data analysis (Lowe, 2004; Cai et al., 2021; Shakhnarovich et al., 2008; Aumüller et al., 2020; Douze et al., 2024).

ANN algorithms attempt to reduce cost in two independent ways. *Sketch*-based techniques (Razenshteyn, 2017; Wang et al., 2014) attempt to compress every point into a short-coded representation, a summary, so that approximate distances can be quickly evaluated. *Index*-based methods (Gani et al., 2016) pre-partition the space, and examine only a subset of the point set, at query time. The two approaches are complementary and often combined in practice. The focus of this work is on indexing methods, and most specifically *space-partitioning* in the $(\mathbb{R}^d, \delta_{\mathcal{E}})$ metric space (Aumüller et al., 2020). The space is divided into cells $\mathcal{B}_g$, each storing data points, and a query touches only those stored in the cell that contains the query point (plus a few neighbors for higher recall).

Space-partition indices are practical and efficient with their small space overhead, each cell stores a representative, e.g., a centroid, and a list of point IDs, far less than e.g., graph indices need (Malkov & Yashunin, 2018). Cells can be queried in parallel by different cores, read in one-shot by GPUs, and fetched as one block (I/O call) from the disk storage (Johnson et al.,

2019; Douze et al., 2024; Jayaram Subramanya et al., 2019). These strengths hold only when the cells are well built, whether by fixed, *data-independent* rules or by partitions *learned* from the data. What drives performance is how cells are built. Data-independent (Andoni & Indyk, 2008; Andoni et al., 2018) rules fix splits a priori (e.g., random hyperplanes, simple trees), so they build in $\mathcal{O}(|X|d)$ time but ignore the geometry of $X$, and recall degrades on clustered or curved regions. *Data-dependent* schemes (e.g., k-means/IVF families (Jegou et al., 2010), learned hashing (Wang et al., 2015)) fit cell parameters to $X$ and usually improve recall per number of candidates visited.

In practice, partitions often tend to be isotropic, for example Voronoi cells around $k$-means centroids (Lloyd, 1982; Arthur & Vassilvitskii, 2007), and a single global number of cells $K$ is chosen. These design choices then cause predictable errors on heavy-tailed data (Clauset et al., 2009). *Partition resolution*, a single global number of cells $K$ applied everywhere, means dense regions get fragmented into many small (near-spherical) cells, while sparse regions are covered by a few large ones (Du et al., 1999). *Balanced partitions* on non-uniform data create a complementary problem: in dense zones they split true neighbors across cells, and in sparse zones they pack unrelated points together to meet the size target (Malinen & Fränti, 2014; Aumüller et al., 2020). Neighborhoods are approximately Euclidean only locally; large cells merge unrelated regions, while overly small ones fragment continuous neighborhoods. To recover locality one must either increase $K$ or probe (touch), many adjacent cells (Johnson et al., 2019; Lv et al., 2007), which raises candidate (distance) counts and hurts low-probe regimes. This leaves a concrete gap: partitions that capture local geometry and adapt their local resolution (effective cell size / expected cell cardinality) to density under a principled objective that balances reconstruction fidelity against candidate count.

**Contributions.** We propose **GARLIC**, a geometry-aware space-partition index for Euclidean ANN, optimized under an information-theoretic objective that balances coverage, overlap, and budget efficiency. Under this objective, GARLIC learns a probabilistic partition of $\mathbb{R}^d$ into Gaussian cells whose shape and placement align with local principal directions and whose sizes adapt to local density, adding capacity only where needed through local adaptive refinement. The resulting partition reduces cross-cell neighbor splits and candidate counts under small probe budgets.

- **Anisotropic, density-adaptive partition.** GARLIC represents $\mathbb{R}^d$ with Gaussian cells that follow local geometry and adapt to density, improving within-cell neighbor cohesion under small candidate budgets. (Section 2.1 – 2.3)

- **Information-theoretic objective.** We balance coverage, overlap, and probe efficiency via expected Mahalanobis coverage, an assignment-entropy penalty, and geometric anchoring regularization. (Section 2.2)

- **Local adaptive refinement.** We add Gaussians only where needed through split/clone operations triggered by cell size and spill ratio, avoiding a single global resolution (one $K$ everywhere). (Section 2.3)

- **Budget-centric evaluation and analysis.** We report competitive performance across multiple accuracy and distortion metrics under candidate and distance budgets on standard Euclidean benchmarks, and provide build/query/space complexity together with ablations that isolate each component's contribution (Section 3, Appendix A.1).

## 1.1 RELATED WORK

**Traditional ANN Families.** ANN methods fall into three main families: (i) *sketching and compression*, which encode vectors into compact codes for fast distance estimation (e.g., product quantization (Jegou et al., 2010), optimized PQ (Ge et al., 2013), iterative quantization (Gong et al., 2012)); (ii) *index-based methods*, which pre-organize the dataset to reduce the number of points touched at query time (e.g., IVF (Jegou et al., 2010), PCA-trees (Sproull, 1991), randomized projections and Johnson–Lindenstrauss-based embeddings (Anagnostopoulos et al., 2018); and (iii) *graph-based methods*, which traverse neighborhood graphs during

search (e.g., HNSW (Malkov & Yashunin, 2018), DiskANN (Jayaram Subramanya et al., 2019)). Within indices, our work focuses on the sub-family of *space-partition indices*, which balance memory efficiency with parallelizability and provide a probe-based complexity model compatible with GPU and IO acceleration.

**Data-Independent Partitions.** Data-independent indices split space according to fixed random rules, ignoring the geometry of the dataset. A canonical example is hyperplane LSH, which assigns points based on the sign of random projections and can be queried more flexibly via multi-probing (Indyk & Motwani, 1998; Lv et al., 2007). These methods offer theoretical guarantees and fast build times, but their isotropic and geometry-agnostic partitions lead to poor recall on clustered or manifold-structured data. GARLIC instead learns anisotropic, density-adaptive cells aligned with the underlying data geometry.

**Data-Dependent Partitions.** Classical learned indices often rely on isotropic partitions with a fixed global number of cells. $k$-Means and its inverted-file variants (IVF) assign points to centroid Voronoi cells (Lloyd, 1982; Jegou et al., 2010), while PCA-trees split recursively along principal components (Sproull, 1991). Scalable extensions include mini-batch k-means (Sculley, 2010), BIRCH, which builds a hierarchical clustering tree with compact representations (Zhang et al., 1996), and BLISS, which incrementally refines partitions for large datasets (Gupta et al., 2022). These methods are efficient, but their isotropic cells and global resolution fragment dense regions and mix unrelated points in sparse ones. Neural LSH takes a different approach by building balanced cuts of the $k$-NN graph and training a classifier to extend them to $\mathbb{R}^d$ (Dong et al., 2020). While this can outperform k-means in some settings, the emphasis on balance rather than geometry often splits true neighbors in dense areas and merges unrelated points in sparse areas, raising candidate counts. Gaussian mixture models (GMMs) capture local covariance through Mahalanobis metrics and soft assignments (Dempster et al., 1977; Banerjee et al., 2005). These models demonstrate the benefits of anisotropy, but they maximize likelihood rather than probe efficiency and do not refine capacity locally. GARLIC combines the strengths of these directions by learning anisotropic, density-adaptive partitions with local split/clone refinement under an information-theoretic probe-budget objective, explicitly tailored to ANN retrieval.

The remainder of this work is organized as follows: Section 2 introduces the GARLIC framework, including Gaussian parameterization, the information-theoretic optimization objective, and adaptive refinement strategies. Section 3 presents our experimental evaluation on standard Euclidean benchmark datasets, a set of crucial ablation studies, and GARLIC's limitations. Finally, conclusions are drawn in Section 4.

## 2 METHOD

GARLIC uses a collection of Gaussians $\mathcal{G} = \{\mathcal{N}(\mu_i, \Sigma_i)\}_i$, whose means and covariances adapt to the underlying data distribution, to partition $X \subset \mathbb{R}^d$ into cells for ANN. We choose Gaussians because their mean $\mu$ and covariance $\Sigma$ jointly encode geometry and statistics: eigenvectors of $\Sigma$ capture local principal directions, eigenvalues control anisotropy and effective dimensionality, and the Mahalanobis distance $\delta_M^2(\mathbf{x}, g_i) = (\mathbf{x} - \boldsymbol{\mu}_i)^\top \boldsymbol{\Sigma}_i^{-1}(\mathbf{x} - \boldsymbol{\mu}_i) = \|\mathbf{L}_i^{-1}(\mathbf{x} - \boldsymbol{\mu}_i)\|_2^2$, is the canonical quadratic form associated with the covariance. Level sets of $\delta_M$ correspond to $\chi_d^2$ probability contours, giving a calibrated notion of coverage. Parameterizing $\boldsymbol{\Sigma}$ via its Cholesky factor yields closed-form (Section 2.1), differentiable gradients while guaranteeing positive definitenes, which makes Gaussians uniquely well-suited among partitioning primitives for end-to-end optimization, unlike boxes or zonotopes that lack smooth probabilistic distance functions. Compared to alternatives such as Gaussian log-likelihood, which adds normalization terms unrelated to retrieval efficiency, or KL divergence, which compares distributions rather than points, Mahalanobis provides a direct, efficient, and anisotropy-aware metric for both training and retrieval.

We train the Gaussians with information-theoretic objectives that balance coverage, assignment confidence, and structural consistency (Section 2.2), and refine capacity only where needed via split and clone operations, avoiding uniform or balanced partitions that fragment dense regions or merge sparse ones (Dong et al., 2020; Gong et al., 2012; Arthur &

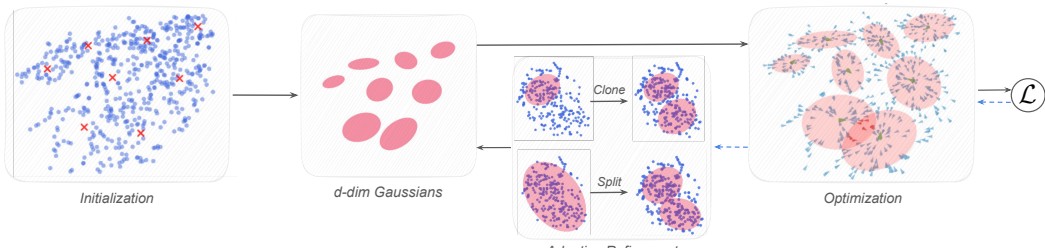

*Initialization*    *d-dim Gaussians*    *Clone*    *Split*    *Optimization*

*Adaptive Refinement*

Figure 1: Overview of GARLIC. Input vectors are represented by Gaussian cells (means, covariances) optimized with information-theoretic objectives for coverage, confidence, and consistency. Cell density is refined adaptively via split/clone operations, and queries use spherical quantization with Mahalanobis distance for cell selection and retrieval. Blue arrows indicate transferring information of gradients (back-propagation) or statistics.

Vassilvitskii, 2006; Kumar et al., 2008; Abdullah et al., 2014; Sproull, 1991; McNames, 2001) (Section 2.3). At query time, each Gaussian cell is equipped with a local hyper-spherical quantization index to narrow candidate searches (Section 2.4), and Mahalanobis distances are again used for cell selection and prioritized bin search (Section 2.5). This approach enables GARLIC to adapt to data geometry while maintaining efficient retrieval, as illustrated in Figure 1.

### 2.1 Gaussian Parameterization

Each Gaussian $g_i$ is defined by its mean $\boldsymbol{\mu_i} \in \mathbb{R}^d$ and covariance $\boldsymbol{\Sigma}_i \in \mathbb{R}^{d \times d}$, parameterized as $\boldsymbol{\Sigma}_i = \mathbf{L}_i \mathbf{L}_i^\top$, $\mathbf{L}_i \in \mathbb{R}^{d \times d}$ being a lower triangular Cholesky factor of matrix $\boldsymbol{\Sigma}_i$, ensuring positive definiteness (a prerequisite for valid Gaussian distributions). The means $\boldsymbol{\mu_i}$ are initialized using K-Means++ (Arthur & Vassilvitskii, 2007) on a subset of the training set. Cholesky factors $\mathbf{L}_i$ are initialized as: $\mathbf{L}_i = \log(\bar{\delta_{\mathcal{E}}}) \cdot \mathbf{I}_d + \boldsymbol{\epsilon}$, where $\mathbf{I}_d \in \mathbb{R}^{d \times d}$ is the identity matrix, and $\bar{\delta_{\mathcal{E}}}$ is the mean Euclidean distance of each $\boldsymbol{\mu_i}$ to its three nearest neighbors. This ensures that the initial scale of each Gaussian reflects the local data density. Perturbation $\boldsymbol{\epsilon} \in \mathbb{R}^{d \times d}$ is a random lower-triangular matrix with entries $\epsilon_{jk} = 2\sigma(r_{jk}) - 1$ if $j > k$ and $0$ otherwise, where $r_{jk} \sim \mathcal{U}(0, 0.01)$ and $\sigma(\cdot)$ is the sigmoid function. This construction yields diagonal-dominant $\mathbf{L}_i$, stabilizing optimization while allowing anisotropic covariances.

### 2.2 Optimization Objective

High-dimensional indexing requires objectives that remain stable under the curse of dimensionality and robust to heterogeneous feature distributions. Classical partitioning heuristics (e.g., balanced splits, uniform clustering) degrade in such regimes: dense regions are fragmented, sparse regions are merged, and distance metrics lose discriminative power Aggarwal et al. (2001); Aumüller et al. (2020). To overcome this, we draw inspiration from self-supervised learning methods such as VICReg Bardes et al. (2022) and Barlow Twins Zbontar et al. (2021), which employ information-theoretic objectives to increase representational capacity without supervision. Analogously, we introduce objectives that guide Gaussians to (i) cover the data distribution, (ii) assign points with high confidence, and (iii) remain anchored to local structure. This replaces heuristic spatial rules with principled, differentiable criteria suited to high-dimensional retrieval.

More specifically, we introduce a divergence-based objective that acts as a reconstruction loss, and regularize it to prevent information explosion (i.e., uncontrolled growth and excessive overlap of Gaussians). The divergence loss $\mathcal{L}_{\text{div}}$ is defined as:

$$\mathcal{L}_{\text{div}} = \frac{1}{N} \sum_{\mathbf{x} \in X} \left( \min_{g_i \in \mathcal{G}} \delta_M(\mathbf{x}, g_i) - \tau \right)^+, \tag{1}$$

where $\delta_M(\mathbf{x}, g_i) = ||\mathbf{L}_i^{-1}(\mathbf{x} - \boldsymbol{\mu}_i)||_2$ denotes the Mahalanobis distance, $(\cdot)^+ = \max(0, \cdot)$ and $\tau$ a standard deviation threshold. It penalizes points that fall outside a Gaussian's coverage

radius ($\delta_M(\mathbf{x}_i, g_i) > \tau$), encouraging Gaussians to expand and cover these points, while points inside ($\delta_M(\mathbf{x}_i, g_i) \leq \tau$) do not contribute, allowing controlled expansion but preventing infinite growth.

Still, Gaussians can overlap, leading to redundant information and performance loss. To surpass this issue, and mitigate fuzzy assignments of points to Gaussians, we introduce a covariance-based regularization, which encourages each Gaussian to dominate its assigned points. Specifically, given a point $\mathbf{x}$, we define its coverage set, i.e., the set of Gaussians that satisfy the coverage radius constraint, as $\mathcal{M}(\mathbf{x}) = \{g_i \in \mathcal{G} \mid \delta_M(\mathbf{x}, g_i) \leq \tau\}$. Then, we compute the normalized soft-assignment probabilities based on Euclidean distances ($\delta_{\mathcal{E}}$) as $p_i(\mathbf{x}) = e^{-\delta_{\mathcal{E}}(\mathbf{x}, \boldsymbol{\mu}_i)} / \sum_{g_j \in \mathcal{M}(\mathbf{x})} e^{-\delta_{\mathcal{E}}(\mathbf{x}, \boldsymbol{\mu}_j)} + \epsilon, \ \forall g_i \in \mathcal{M}(\mathbf{x})$. The covariance loss $\mathcal{L}_{\text{cov}}$ is defined as:

$$\mathcal{L}_{\text{cov}} = 1 - \frac{1}{N} \sum_{\mathbf{x} \in X} \max_{g_i \in \mathcal{M}(\mathbf{x})} p_i(\mathbf{x}) \tag{2}$$

This loss encourages highly confident (low-entropy) assignments, thereby reducing ambiguity and stabilizing optimization.

To further prevent excessive expansion of Gaussians caused by the divergence objective and ensure that each Gaussian aligns with its assigned points, we introduce the anchor loss $\mathcal{L}_{\text{anchor}}$:

$$\mathcal{L}_{\text{anchor}} = \frac{1}{d|\mathcal{G}|} \sum_{g_i \in \mathcal{G}} \left( ||\boldsymbol{\mu}_i - \hat{\boldsymbol{\mu}}_i||_2^2 + \alpha ||\mathbf{L}_i \mathbf{L}_i^\top - \hat{\boldsymbol{\Sigma}}_i||_F^2 \right), \tag{3}$$

where $\hat{\boldsymbol{\mu}}_i$ and $\hat{\boldsymbol{\Sigma}}_i = \text{Cov}(x \in \mathcal{B}_{g_i})$ are the empirical mean and covariance of points assigned to Gaussian $g_i$, and $\alpha$ is a hyperparameter balancing position and shape. This loss constrains Gaussians by anchoring them closely to their local data distributions, restraining uncontrolled growth from other loss terms and maintaining geometric alignment. Finally, our loss function is defined as:

$$\mathcal{L} = \lambda_{\text{div}} \cdot \mathcal{L}_{\text{div}} + \lambda_{\text{cov}} \cdot \mathcal{L}_{\text{cov}} + \lambda_{\text{anchor}} \cdot \mathcal{L}_{\text{anchor}},$$

where $\lambda_{\text{div}}, \lambda_{\text{cov}}$ and $\lambda_{\text{anchor}}$ are hyperparameters balancing the importance of each term.

### 2.3 Adaptive Refinement

Controlling the number of Gaussians during training is key for adapting to data complexity; allocating more in dense regions and fewer in simpler ones. We employ an adaptive refinement strategy that adjusts the Gaussian set based on local point density. Prior approaches rely on positional gradients to guide refinement (Kerbl et al., 2023), but these become sparse and unreliable in high dimensions. In our case, we

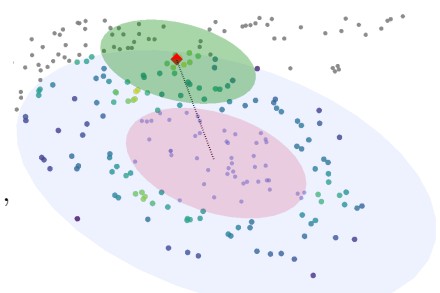

(a) Clone operation. (light red) region where $d_M(x, g_i) \leq \tau$; (blue) outer shell $\tau \leq d_M(x, g_i) \leq e\tau$; (gray) $\{x : d_M(x, g_i) \geq e\tau\}$; (red) mean of the new gaussian; (green) new covariance matrix. (purple - lime) denote increasing data density.

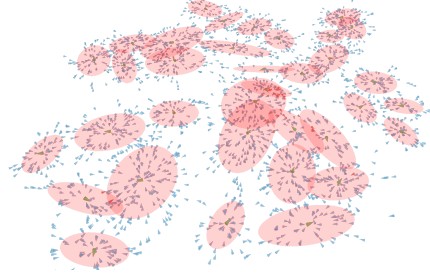

(b) Grad flow: $\mathcal{L}_{\text{div}}, \mathcal{L}_{\text{cov}}, \mathcal{L}_{\text{anchor}}$

Figure 2: Gaussian refinement and associated loss gradients.

also require search efficiency, so we refine Gaussians whose cell cardinality exceeds a threshold: $|\mathcal{B}_g| > \gamma \cdot |X|$. *Splitting* is performed by applying clustering (Arthur & Vassilvitskii, 2007) to produce two means $\boldsymbol{\mu}_1, \boldsymbol{\mu}_2$, while covariances are scaled down as $\mathbf{L}_1 = \mathbf{L}_2 = \alpha \cdot \mathbf{L}$, with $\alpha < 1$.

For *cloning*, we target Gaussians with a high ratio of outside points to interior ones $\mathcal{B}_{g_i}^{in} = \{\mathbf{x} \in X : \delta_M(\mathbf{x}, g_i) \leq \tau\}$. Instead of using all nearest outside points, we focus on a

boundary region, and select $k = \rho \cdot |\mathcal{B}_g^{out}|$ points, where $\rho \in (0,1)$ is a sampling ratio and $\mathcal{B}_{g_i}^{out} = \{\mathbf{x} \in X : \tau < \delta_M(\mathbf{x}, g_i) < e\tau \text{ and } g_i = \arg\min_{g_j \in \mathcal{G}} \delta_M(\mathbf{x}, g_j)\}$ with $e > 1$ is the set of all boundary points. The sampled subset $S = \{\mathbf{x}_1, \mathbf{x}_2, ..., \mathbf{x}_k\} \subset \mathcal{B}_g^{out}$ is chosen randomly without replacement. This sampling approach serves several purposes, reducing computational cost, as density estimation in high dimensions is expensive, providing statistical robustness by focusing on representative points and helping to avoid outliers that might exist in the boundary region as Figure 2 showcases. The candidate Gaussians are picked with regard to the ratio of boundary to interior points: $|\mathcal{B}_g^{out}|/|\mathcal{B}_g^{in}| > \beta$. The center of the new Gaussian is placed at the point with the highest local density in the boundary region, such that for a point $\mathbf{p}$ in the boundary region, we compute its local density as the inverse mean Euclidean distance from the 3-NN and select the point with the highest density: $\mathbf{p}^* = \arg\max_{\mathbf{p}} \rho(\mathbf{p})$. The covariance matrix is then cloned, with $\mathbf{L}_{new} = \mathbf{L}$. In our method, splitting is prioritized over cloning: when a Gaussian grows too large, we first partition it to reduce cell cardinality and maintain search efficiency. Cloning is applied only if splitting is not triggered, serving to refine coverage near dense boundary regions without inflating candidate counts unnecessarily.

## 2.4 Quantization

After optimizing our Gaussians, we assign points that fall outside the coverage radius to the nearest Gaussian such that our final cells are: $\mathcal{B}_{g_i} = \{\mathbf{x} \in X \mid \delta_M(\mathbf{x}, g_i) \leq \tau\} \cup \{\mathbf{x} \in X \mid \delta_M(\mathbf{x}, g_i) > \tau \text{ and } g_i = \arg\min_{g_j \in G} \delta_M(\mathbf{x}, g_j)\}$. While Gaussian cells provide an effective partition of the space, brute-force search within them is still prohibitive in high dimensions. Standard ANN methods usually quantize globally, ignoring the anisotropy and local geometry already captured by our Gaussians. We instead introduce a localized quantization scheme: once points are assigned to cells, each cell is treated in its own coordinate system, where Euclidean structure is better aligned with the underlying data. Quantizing in this local basis reduces the number of distance computations per query, while preserving the geometry captured by Mahalanobis distance.

For each cell $\mathcal{B}_g$, we apply PCA to reduce dimensionality while preserving the local structure $\mathbf{P}_g = \text{PCA}(\{\mathbf{x} - \bar{\mathbf{x}}_g \mid \mathbf{x} \in \mathcal{B}_g\}, r)$ where $\bar{\mathbf{x}}_g$ is the mean of points in cell $\mathcal{B}_g$, $r$ is the reduced dimensionality (typically $r \ll d$, constant in practice), and $\mathbf{P}_g \in \mathbb{R}^{d \times r}$ contains the top-$r$ principal components. This step both lowers computational cost and aligns the local coordinate system with the main variance directions of the data. Each point $\mathbf{x} \in \mathcal{B}_g$ is then projected into this subspace $\mathbf{x}^r = \mathbf{P}_g^\top (\mathbf{x} - \bar{\mathbf{x}}_g)$, which embeds the data in $\mathbb{R}^r$ while preserving Euclidean structure up to the discarded components. Finally, we convert $\mathbf{x}^r$ into hyperspherical coordinates $\mathbf{s} = \text{cart2sph}(\mathbf{x}^r) = (s_1, s_2, \ldots, s_r)$, where $s_1$ denotes the radial component $\|\mathbf{x}^r\|_2$ and $s_2, \ldots, s_r$ are angular coordinates. This reparameterization retains the Euclidean metric but enables partitioning along radial and angular directions.

The hyperspherical space is partitioned into bins $\mathcal{Q}_g = \{B_{i,\mathbf{j}} \mid i \in \{1, \ldots, n_r\}, \mathbf{j} \in \{1, \ldots, n_a\}^{r-1}\}$, with radial boundaries $r_i = r_{\min} + (r_{\max} - r_{\min})\frac{i}{n_r}$, $i = 0, \ldots, n_r$, and angular boundaries $\theta_{j,k} = \theta_{\min,k} + (\theta_{\max,k} - \theta_{\min,k})\frac{j}{n_a}$, $j = 0, \ldots, n_a$, $k = 1, \ldots, r-1$. The dominant cost of index construction comes from full Mahalanobis-based assignments during optimization, requiring $\mathcal{O}(|X| \cdot K \cdot d^2)$ time per iteration, while initialization, refinement, and PCA quantization add only lower-order terms (see Appendix A.3).

## 2.5 Inference

Given a query $\mathbf{q}$, we first select the top-$k_G$ Gaussians by Mahalanobis score $\delta_M(\mathbf{q}, g)$. For each selected Gaussian $g$, we project $\mathbf{q}$ to its local PCA space $\mathbf{q}_g^r = \mathbf{P}_g^\top (\mathbf{q} - \bar{\mathbf{x}}_g)$ and convert to hyperspherical coordinates $\mathbf{s}_g = \text{cart2sph}(\mathbf{q}_g^r)$. We prioritize bins by a query-to-bin distance computed in the reduced Euclidean space. Each bin $B_{i,\mathbf{j}}$ is defined by spherical bounds $r \in [r_i, r_{i+1}]$ and $\theta_k \in [\theta_{j_k,k}, \theta_{j_k+1,k}]$ for $k = 1, \ldots, r-1$. We define:

$$d(\mathbf{q}_g^r, B_{i,\mathbf{j}}) = \min_{\boldsymbol{\phi} \in \mathbb{R}^r} \left\|\mathbf{q}_g^r - \text{sph2cart}(\boldsymbol{\phi})\right\|_2 \text{ s.t. } r_i \leq \phi_1 \leq r_{i+1}, \ \theta_{j_k,k} \leq \phi_{k+1} \leq \theta_{j_k+1,k}.$$

If $\mathbf{s}_g$ lies inside $B_{i,\mathbf{j}}$, then $d(\mathbf{q}_g^r, B_{i,\mathbf{j}}) = 0$; otherwise we solve the bound-constrained problem (Byrd et al., 1995) initialized at $\phi_1 = \text{clip}(\|\mathbf{q}_g^r\|_2, [r_i, r_{i+1}])$ and $\phi_{k+1} = 0.5(\theta_{j_k,k} + \theta_{j_{k+1},k})$. Bins in $\mathcal{Q}_g$ are sorted by $d(\mathbf{q}_g^r, \cdot)$ ascending and scanned until a bin budget $\rho \in (0, 1]$ of bins per cell is exhausted (typically $\rho = 0.3$). Within each visited bin we compute exact Euclidean distances in the original space between $\mathbf{q}$ and all points indexed in that bin, aggregating candidates across the $k_G$ selected Gaussians. The final result is obtained by selecting the overall nearest neighbors among the accumulated candidates, with the dominant inference cost coming from distance computations (see Appendix A.3).

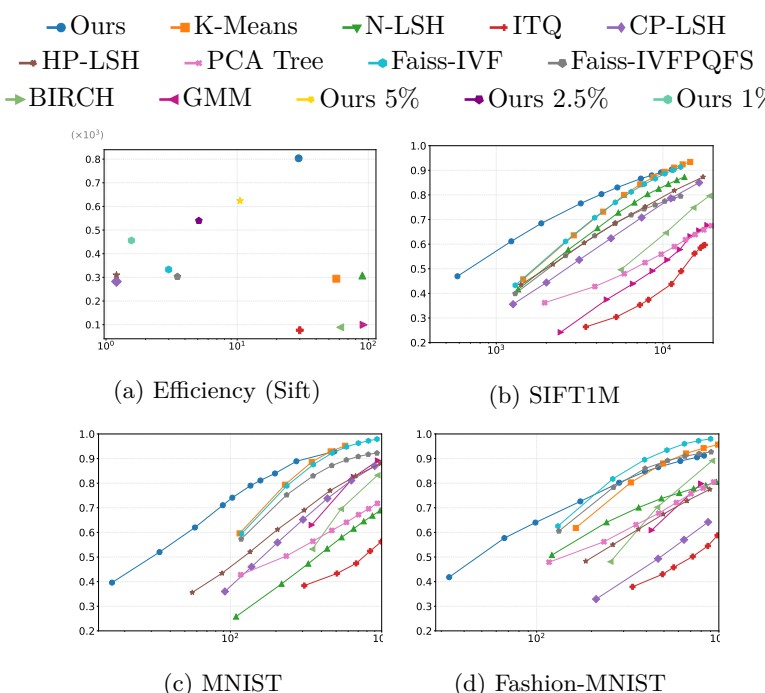

(a) Efficiency (Sift)     (b) SIFT1M

(c) MNIST      (d) Fashion-MNIST

Figure 3: Efficiency and retrieval performance across datasets, measured via Recall@1. Methods closer to the top-left (↖) indicate a better trade-off between accuracy and candidate count. GARLIC performs especially well in the low-probe regime. Our method is trained in 10% for SIFT-1M unless stated otherwise.

## 3 EXPERIMENTS

**Datasets.** We evaluate our method on three benchmark datasets: `SIFT1M` (Lowe, 2004) (128-dimensional image descriptors with one million points), `MNIST` (LeCun et al., 1998) and `Fashion-MNIST` (Xiao et al., 2017) (784-dimensional vectors from $28 \times 28$ grayscale images) following the standard setup in ANN-Benchmarks (Aumüller et al., 2020). Further information about the datasets can be found in the Appendix.

**Evaluation.** We evaluate GARLIC on the approximate nearest neighbor task, reporting Recall@1 and as function of the number of distance computations while varying the probe budget. Recall@1 measures the accuracy of the very first retrieved neighbor, reflecting ranking performance, and is strict. In Appendix A.1, we showcase more related metrics such as $\epsilon$-Recall, empricial c-approximation factors and mean relative error. We use distance computations, rather than wall-clock latency (QPS), as the efficiency axis. Distance computations capture the algorithmic effort of a search procedure independently of hardware and implementation details (different CPUs/GPUs, BLAS kernels, batching, I/O) (Aumüller et al., 2020; Peng et al., 2023). This metric directly reflects the goal of space-partitioning methods, which is to minimize the number of distances required to achieve a target recall.

**Baselines and Comparisons.** We evaluate GARLIC against representative approximate nearest neighbor methods such as $k$-Means++ (Arthur & Vassilvitskii, 2006), BIRCH (Zhang

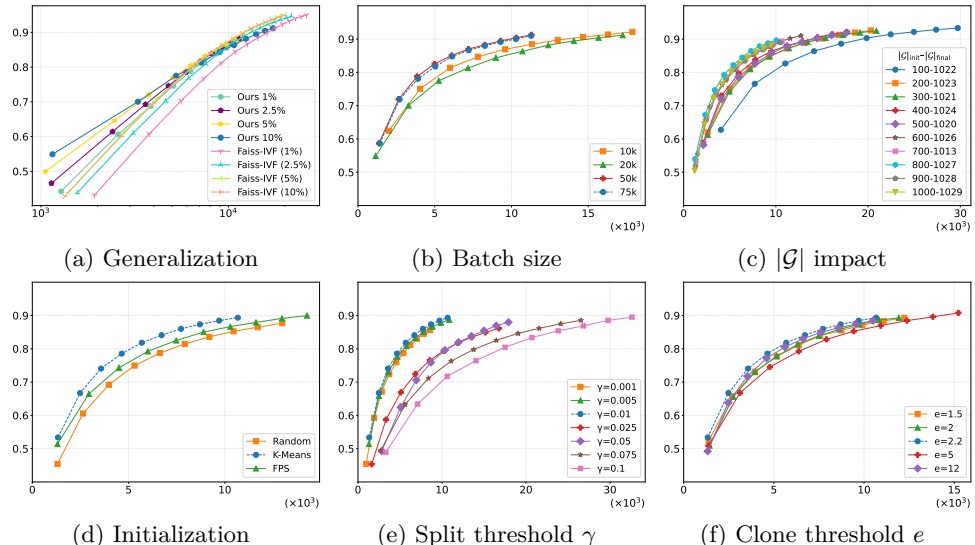

Figure 4: (a) Generalization with limited training data. (b) Batch size effect. (c) Impact of initial number of Gaussians. (d) Effect of different initialization strategies. (e) Impact of split threshold $\gamma$, with higher values increasing accuracy at the cost of more probes. (f) Effect of clone threshold $e$. The blue dashed regards the parameters used in main experiments, and top-left is better ($\nwarrow$), while all y-axis report Recall@1 and x-axis candidate counts.

et al., 1996), Gaussian Mixture Models (GMM) (Dempster et al., 1977; Banerjee et al., 2005), Neural-LSH (N-LSH) (Dong et al., 2020), Hyperplane LSH (HP-LSH) (Indyk & Motwani, 1998), Cross-polytope LSH (CP-LSH) (Andoni et al., 2015), ITQ (Gong et al., 2012), PCA Tree (Kumar et al., 2008; Abdullah et al., 2014; Sproull, 1991; McNames, 2001), Faiss-IVF and Faiss-IVFPQFS (Douze et al., 2024). All baselines were run under identical conditions, using the standardized ANN-Benchmarks datasets and query splits, with hyperparameters optimized according to their respective publications and usage.

**Results and Analysis.** We present performance curves across datasets, reporting Recall@1 as a function of the number of retrieved candidates. As shown in Figure 3, our method performrs efficiently with respect to other baselines. GARLIC generally achieves preferable recall–efficiency trade-offs than traditional space partitioning methods, including inverted-file indices (Faiss-IVF, IVFPQFS), hashing approaches (HP-LSH, CP-LSH, Neural-LSH), and tree-based methods (PCA-Tree, BIRCH). Figure 3(a) evaluates efficiency through Recall@1-per-probe versus build time. All GARLIC variants occupy the upper-left region, indicating favorable trade-offs: lightweight models trained with 1% of the data already surpass strong baselines, while larger-capacity variants (e.g., 10% training) further improve recall without excessive cost. In contrast, competing methods cluster in lower-efficiency or higher-cost regions, reflecting less balanced trade-offs between indexing overhead and search quality.

Table 1: Effect of loss terms and adaptive refinement average Recall@1 / Candidates ($\times 10^5$). Higher is better.

| Configuration | Performance ↑ |
|---|---|
| Loss terms | |
| w/ all | 16.20 |
| $\mathcal{L}_{div} + \mathcal{L}_{cov}$ | 15.38 |
| $\mathcal{L}_{div} + \mathcal{L}_{anchor}$ | 10.64 |
| $\mathcal{L}_{div}$ | 9.78 |
| Split & Clone | |
| w/ both | 16.20 |
| w/ split | 14.22 |
| w/ clone | 5.45 |
| w/o any | 4.63 |

**Ablation study.** We conduct ablation studies to isolate the effect of key design choices and components. Unless stated otherwise, experiments are on SIFT-1M with a training subset of 5% of $|X|$, and Recall@1 is measured against distance computations. We investigate the robustness and data efficiency of our method through controlled downsampling of the training set. As shown in Figure 4(a), our method consistently maintains strong performance across varying training sizes. Notably, even with only 1% of the training data, our model outperforms the Faiss-IVF variants of up to 10% training size in terms of Recall@1 versus

the number of retrieved candidates. This trend remains consistent across higher percentages, demonstrating that our approach learns a compact yet highly effective representation of the data distribution. Furthermore, Figure 4(b) highlights the batch-size effect (here using a fixed training subset of $100k$ points instead of $50k$), where larger batches yield consistently better performance. Finally, Figure 4(c) shows that more initial Gaussians produce better results at equal iterations, reflecting faster convergence. This happens because adaptive density control targets spatial placement rather than Gaussian count.

We next evaluate core design hyperparameters. In Figure 4(d), we investigate initialization strategies for Gaussian centers, revealing that K-Means initialization consistently outperforms alternatives. Although farthest point sampling (FPS) achieves comparable recall, it requires approximately 50% more probes, and catching up to K-Means requires more training time. In Figure 4(e), the split threshold $\gamma$, controlling the density condition for Gaussian subdivision, is examined. The optimal values are $\gamma = 0.005$ and $\gamma = 0.01$, though $\gamma = 0.005$ requires more time due to more frequent splits. As for the clone threshold $e$, depicted in Figure 4(f), which determines the outer shell for cloning, it remains stable across different values, with $e = 2.2$ being preferable in situations requiring high recall.

Table 1 analyzes the impact of loss terms, adaptive refinement and covariance configurations. Split and clone are pivotal for performance, while the covariance term ($\mathcal{L}_{\text{cov}}$) significantly boosts retrieval. The anchor term ($\mathcal{L}_{\text{anchor}}$) has minor effect on raw performance but is essential for making Gaussians geometrically informative. Additional ablations and results are provided in the Appendix.

**Limitations and Future Work.** GARLIC employs Mahalanobis distance and Gaussian primitives, which assume a Euclidean metric and are therefore not directly compatible with angular similarity. This may be addressed by adopting angular counterparts, such as von Mises–Fisher distributions, which we leave for future work. The trade-off between recall and latency can be further enhanced by augmenting the number of Gaussians and organizing them within tree structures (e.g., KD-tree or Ball-tree over Gaussian means), reducing query cost. The spatial complexity scales quadratically with dimensionality ($\mathcal{O}(Kd^2)$) due to full anisotropic covariances; this challenge can be mitigated through low-rank approximations or quantization of Cholesky factors. Finally, while the method shows some sensitivity to initialization, this is alleviated by the information-theoretic objectives and the progressive refinement via split and cloning. Beyond these limitations, GARLIC is naturally incremental: each Gaussian cell maintains sufficient statistics (mean, covariance, cardinality), allowing new data to be integrated through online updates. Combined with local split/clone refinement, this enables streaming and online learning scenarios without rebuilding the entire index. We leave a systematic evaluation of this capability to future work.

## 4 Conclusions

We introduced GARLIC, a geometric structure that learns the underlying distribution for both approximate nearest neighbor search and classification. By combining information-theoretic objectives with adaptive refinement (split and clone), and representing the space via anisotropic Gaussians, our method achieves competitive performance in Euclidean approximate nearest neighbor, particularly in low-probe regimes. Experiments demonstrate competitive recall-efficiency tradeoffs and robustness under severe data reduction, highlighting its generalization capabilities.

## Statements

**Ethics Statement.** Our method is evaluated exclusively on publicly available benchmark datasets. These datasets contain no personally identifiable or sensitive information, and their licenses are respected. To the best of our knowledge, the proposed method does not raise ethical concerns. We adhere to the ICLR Code of Ethics and Code of Conduct.

**Reproducibility Statement.** We provide detailed descriptions of our initialization, optimization, refinement strategies, and hyperparameters in the main text and Appendix A. All

datasets are standard and publicly available. We will release the full source code, trained models, and experiment scripts upon publication to facilitate reproducibility and support future research.

**LLM Usage.** Large language models (LLMs) were used to assist in editing and rephrasing parts of the manuscript for clarity, and to accelerate the creation of visualizations (e.g., diagnostic figures). All technical contributions, algorithms, and experiments were designed, implemented, and validated by the authors.

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

## A APPENDIX

This appendix provides additional details and experiments as mentioned in the main paper. Section A.1 provides additional results and ablation studies to the ones presented in Section 3 of the main paper. Section A.1.1 contains visualization of diagnostics and statistics regarding our proposed method, to provide a comprehensive understanding of its behavior. Section A.2 contains additional technical details regarding the datasets used and the experimental setup, as well as the training pipeline. Section A.3 discusses and provides detailed computational time and space complexity for the build and query procedures of GARLIC.

### A.1 FURTHER RESULTS

We extend the analysis of the experimental section, by further examining the impact of individual design choices in GARLIC and providing results. Unless noted otherwise, for each result presented as abblation, we sample a training set of 50.000 (5%) from SIFT-1M and only change the requested parameters while keeping all others as is, while reporting recall@1 against distance computations. In each figure, the parameters used in the main experiments are represented by the method indicated by the blue dashed line, and optimal outcomes are achieved when positioned on the top left of the figures ($\nwarrow$). When comparing, for eadability, we choose to exclude GMM, Neural-LSH, and ITQ for their inconsistent and unstable performance.

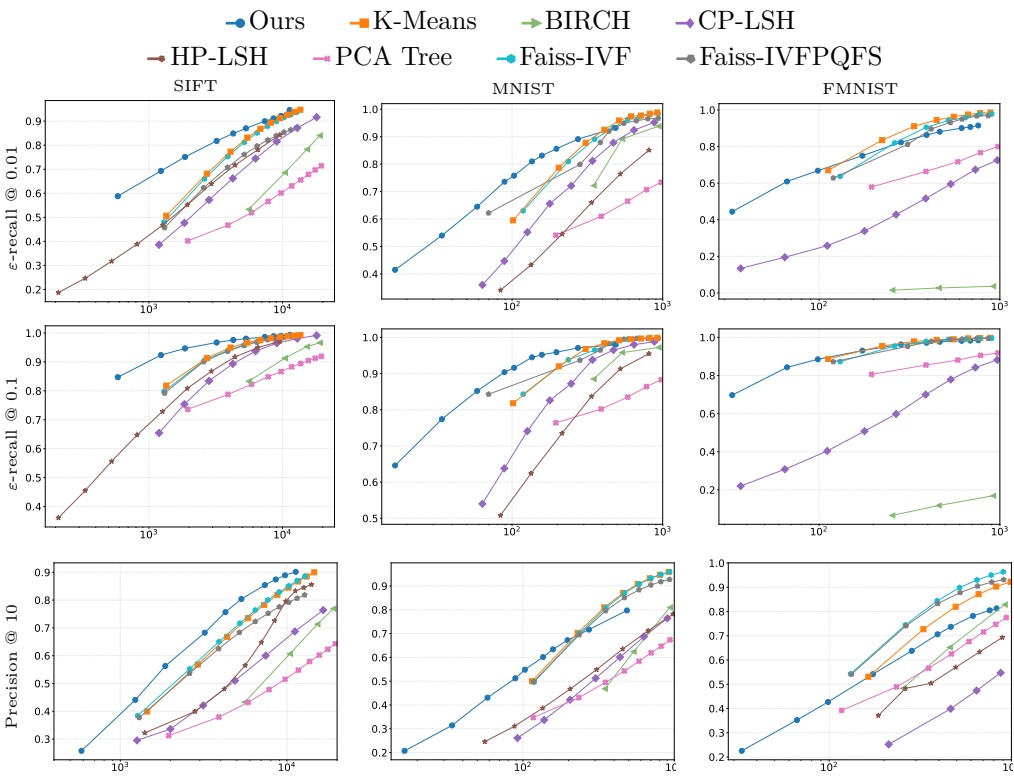

Figure 5: Additional distortion-aware accuracy results (**higher is better**). Each curve reports performance as a function of the candidate budget (x-axis). Panels show $\varepsilon$-Recall at $\varepsilon \in \{0.01, 0.10\}$ and P@10 across SIFT1M, MNIST, and FMNIST. Methods closer to the top-left ($\nwarrow$) are more accurate under smaller candidate budgets.

We evaluate retrieval quality using both accuracy- and distortion-based criteria, as shown in Figures 5 and 6. Beyond Recall@1, we report Precision@10 (P@10), capturing the fraction of retrieved points among the top-10 that are true neighbors. To assess approximation tightness for the nearest neighbor, we measure the distance ratio between the returned neighbor and

the exact nearest neighbor. From this we derive: (i) $\varepsilon$-Recall, the fraction of queries where the retrieved distance is within a factor $(1 + \varepsilon)$ of the ground-truth (with $\varepsilon \in \{0.01, 0.10\}$); (ii) the percentiles $r95$ and $r99$, which correspond to empirical $c$-approximation factors at the 95th and 99th percentiles; and (iii) the mean relative error, $RE_{\mathrm{mean}}$, summarizing the average stretch beyond the true nearest-neighbor distance. These metrics complement recall by quantifying how close the returned distances are to the exact nearest neighbor, and follow standard practice in ANN-Benchmarks.

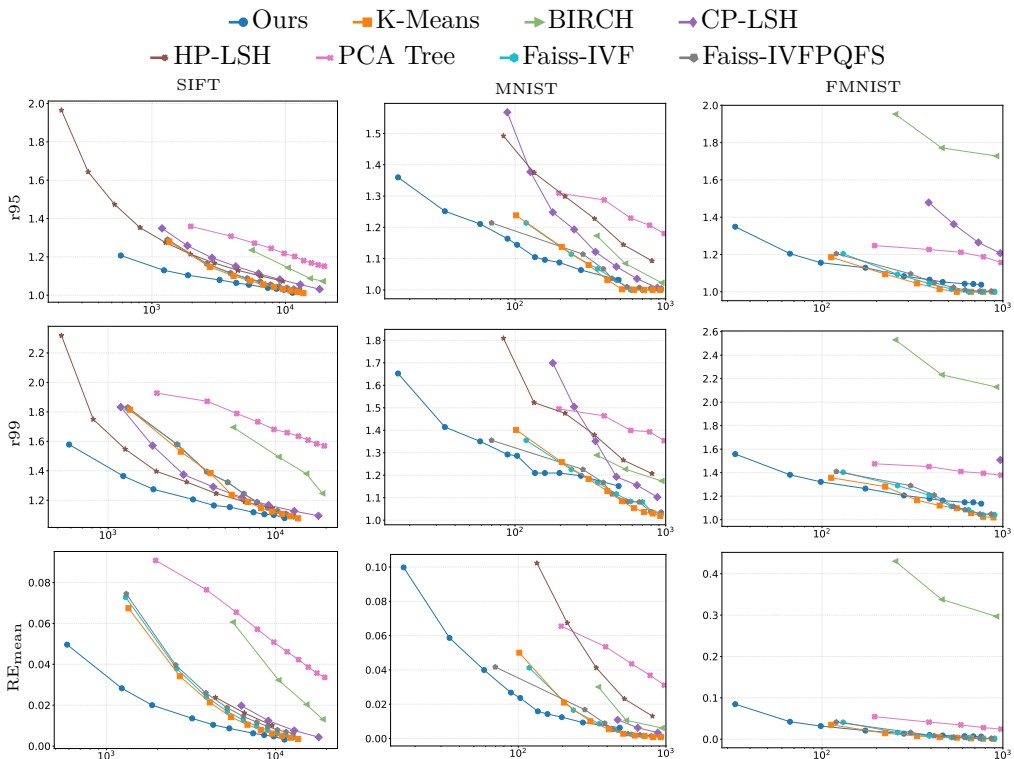

Figure 6: Approximation tightness summaries (**lower is better**). We report the 95th and 99th percentiles of the distance ratio ($r95$, $r99$) and the mean relative error ($RE_{\mathrm{mean}} = \mathbb{E}[r-1]$) versus the candidate budget. Methods closer to the bottom-left ($\swarrow$) achieve tighter approximations with fewer candidates.

Across datasets we observe consistent trends: $\varepsilon$-Recall at 1% and 10% rapidly approaches 1.0, indicating that even under small probe budgets the vast majority of retrieved neighbors lie within 1%–10% of the true nearest-neighbor distance. The $r95$ and $r99$ curves remain close to 1.0, showing that 95%–99% of queries admit near-exact retrieval, while $RE_{\mathrm{mean}}$ stays low and stable, confirming that average distortion is minimal. Together with high P@10, these results demonstrate that the learned Gaussian partitions not only achieve strong recall but also return candidates that are quantitatively close to the exact nearest neighbors, providing both efficiency and fidelity in the ANN process.

**Additional Abblation Studies** In Figure 7(a), the contribution of each loss term to the optimization process is analyzed. The results indicate that the covariance loss $\mathcal{L}_{\mathrm{cov}}$ produces the most significant performance improvement, while the anchor loss $\mathcal{L}_{\mathrm{anchor}}$ functions as an effective regularizer, grounding each Gaussian by aligning it with its corresponding point distribution. Figure 7(b) demonstrates the effects of our adaptive refinement operations, where the split and clone mechanisms improve retrieval performance when used together, each contributing by providing complementary benefits to the quality of representation. Figure 7(c), addresses the parameter $\tau$ within $\mathcal{L}_{div}$, demonstrating that $\tau = 3$ is the optimal choice, particularly in scenarios involving a limited number of probes. Values surpassing

those presented in our study ($\tau > 3$) result in less optimal outcomes. For instance, $\tau = 4$ demonstrates a Recall@1 value of 0.67 when evaluated with 154264 probes.

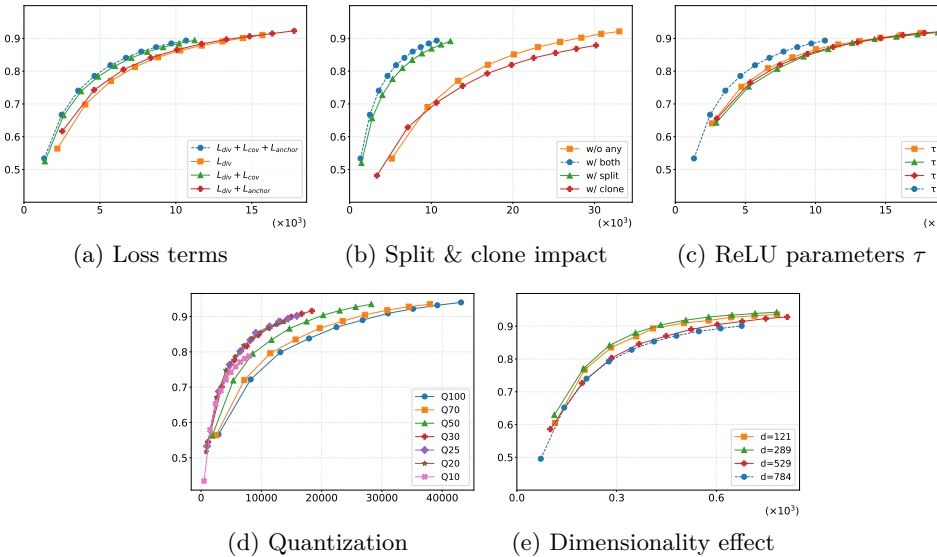

(a) Loss terms  (b) Split & clone impact  (c) ReLU parameters $\tau$

(d) Quantization  (e) Dimensionality effect

Figure 7: Parameter ablations: (a) Impact of different loss term combinations on the Recall-Probe tradeoff, showing that the full loss ($L_{div} + L_{cov} + L_{anchor}$) provides the best balance. (b) Effect of split and clone operations, demonstrating that these operations improve efficiency while maintaining accuracy. (c) Impact of ReLU parameter $\tau$ in the divergence loss. (d) Effect of embedding dimensionality, showing GARLIC's robustness across different dimensions. (e) Efficient search via partial cell scanning. The blue dashed method regards the parameters used in main experiments, and top-left is better ($\nwarrow$).

Furthermore, Figure 7(d) demonstrates the impact of our quantization scheme. In particular, it exhibits strong performance across a broad quantization range ($20-100\%$), with only the most aggressive setting ($10\%$) leading to degradation. This indicates that our quantization strategy is robust and well-aligned with our model structure. Finally, Figure 7(e) examines dimensionality effects using Fashion-MNIST data resized to various dimensions (to simulate higher-dimensional embeddings). The results confirm that GARLIC maintains strong performance across a wide range of dimensionalities.

**Covariance structure vs performance.**
We conduct a targeted study to assess the impact of different covariance configurations-namely, full (anisotropic), diagonal, and isotropic-affect the performance of GARLIC. As shown in Table 2, reducing the Gaussian expressiveness from full to diagonal and then to isotropic leads to a notable decline in average Recall@1 per probe. While diagonal and isotropic configurations reduce both the parameter count and computation per Gaussian to a linear level, $\mathcal{O}(k \cdot d)$, they lead to aggressive pruning and an increased number

Table 2: Effect of anisotropy on average Recall@1 / Probe ($\times 10^5$). Higher is better. As anisotropy decreases, performance degrades due to excessive probe usage.

| Configuration | Performance $\uparrow$ |
|---|---|
| Covariance structure | |
| Anisotropic | 16.20 |
| Diagonal | 1.22 |
| Isotropic | 0.75 |

of probes to make up for the representational loss. This suggests that space complexity cannot be drastically reduced without harming retrieval quality because simpler Gaussian parameterizations result in degraded locality and coverage.

### A.1.1 QUALITATIVE RESULTS

To gain a more comprehensive understanding of the model's behavior, we present a collection of diagnostic visualizations applied to the Fashion-MNIST, MNIST, and SIFT datasets. These plots illustrate structural characteristics, including local coverage, reconstruction fidelity, density patterns, and curvature statistics of the learned Gaussian components. All visualizations are conducted on a randomly sampled subset of training points, utilizing the learned parameters independently of test data supervision.

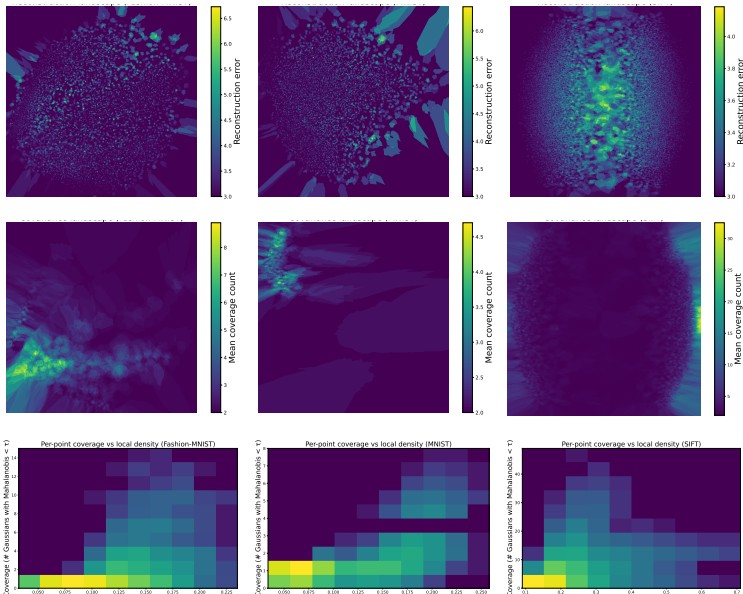

Figure 8: Diagnostic visualizations across Fashion-MNIST (left), MNIST (middle), and SIFT (right). Top row: minimum Mahalanobis reconstruction error; middle row: average Gaussian coverage per point; bottom row: relationship between local density and Gaussian coverage. Bottom row's colormap depicts frequency-density, in logarithmic scale.

**Coverage and reconstruction diagnostics.** Figure 8 illustrates three diagnostic views, each calculated for a different dataset (Fashion-MNIST, MNIST, and SIFT), to evaluate the accuracy of the Gaussian models in representing the datasets. The **top row** includes the *reconstruction landscape*, which visualizes the minimum Mahalanobis distance from each point to any Gaussian. For each dataset, we employ PCA to project the points onto a two-dimensional space and calculate the average reconstruction error of proximate points on a grid. This heatmap highlights how well the Gaussian shells approximate the distribution of data throughout the space.

The **middle row** shows the *coverage landscape*, which counts how many Gaussians fall within the Mahalanobis threshold $\tau$ for each data point. Coverage is computed per point, projected to 2D, and smoothed via k-NN averaging over a grid. This plot reflects the redundancy and spatial spread of the Gaussian coverage. We observe that areas with low reconstruction error tend to be have high coverage.

The **bottom row** depicts the *relationship between local density and Gaussian coverage*. For each point, we compute its local density via the inverse mean distance to its 10 nearest neighbors and correlate this with its coverage count. The resulting 2D histograms reveal structural patterns where regions of higher density generally exhibit greater coverage while sparse regions receive fewer assignments. This aligns with our goal of achieving balanced coverage while maintaining good reconstruction fidelity.

**Curvature-based diagnostics.** Figure 9 provides four views exploring the relationship between curvature and structural properties of the Gaussian assignments across datasets, to

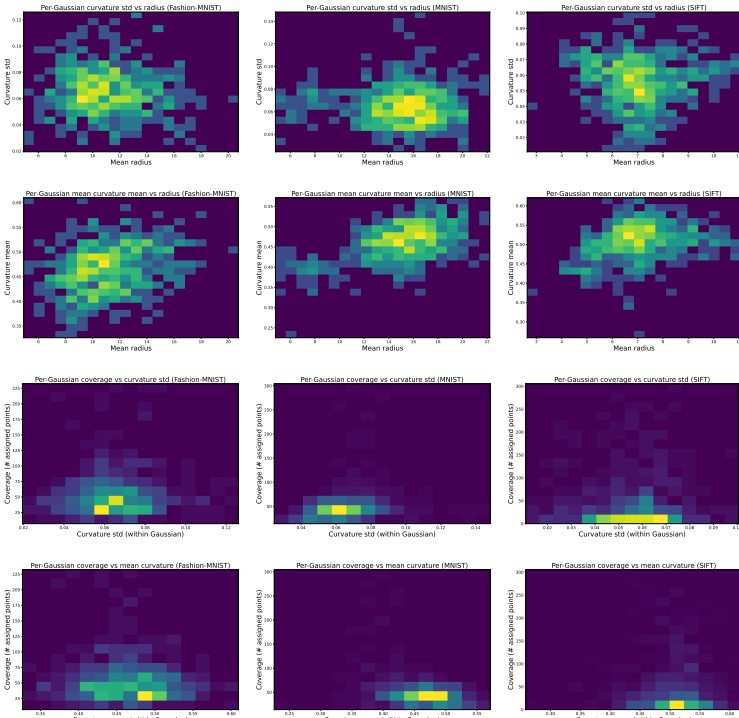

Figure 9: Curvature diagnostics across Fashion-MNIST (left), MNIST (middle), and SIFT (right). Row 1: standard deviation of local curvature vs radius; Row 2: mean local curvature vs radius; Row 3: Gaussian coverage vs curvature std; Row 4: Gaussian coverage vs curvature mean. Coloring accounts for density.

examine wheather the learned model is informative and geometrically consistent with the data $x \in X$.

The **first row** shows the standard deviation of local curvature as a function of the average radius ($l_2$) per Gaussian. For each Gaussian, we collect nearby assigned points (under Mahalanobis threshold $\tau$), compute their curvature via PCA-based local flatness, and report the standard deviation. Each bin aggregates Gaussians by radius and variation in curvature, highlighting the stability of their local geometry, where for each dataset curvatures deviations tend to be around $\sim 0.06$.

The **second row** reports the mean curvature of each Gaussian against its average radius. This indicates the intrinsic dimensionality and shape complexity of regions assigned to Gaussians of different spatial extent. In general, we see that mean curvature tends to be $\sim 0.5$, suggesting a moderate level of local non-linearity, especially for Gaussians with smaller support. As the radius increases, curvature remains relatively stable, indicating consistent local geometry across scales.

The **third row** depicts how Gaussian coverage (number of assigned points) varies with the curvature standard deviation of the assigned region. We observe that most Gaussians exhibit low curvature variability (std $\sim 0.06$), indicating that points within each Gaussian tend to have similar geometric structure. Moreover, there is no clear correlation between coverage and curvature std, implying that heavily used Gaussians are not more geometrically diverse than others. This suggests a form of balanced representation capacity across Gaussians.

The **fourth row** shows coverage as a function of mean curvature, where Gaussians with high coverage have curvature patterns similar to those with low coverage.

Figure 10 demonstrates the capability of GARLIC to capture the intrinsic geometric structure of high-dimensional data with locally varying dimensionality. The anisotropy histograms

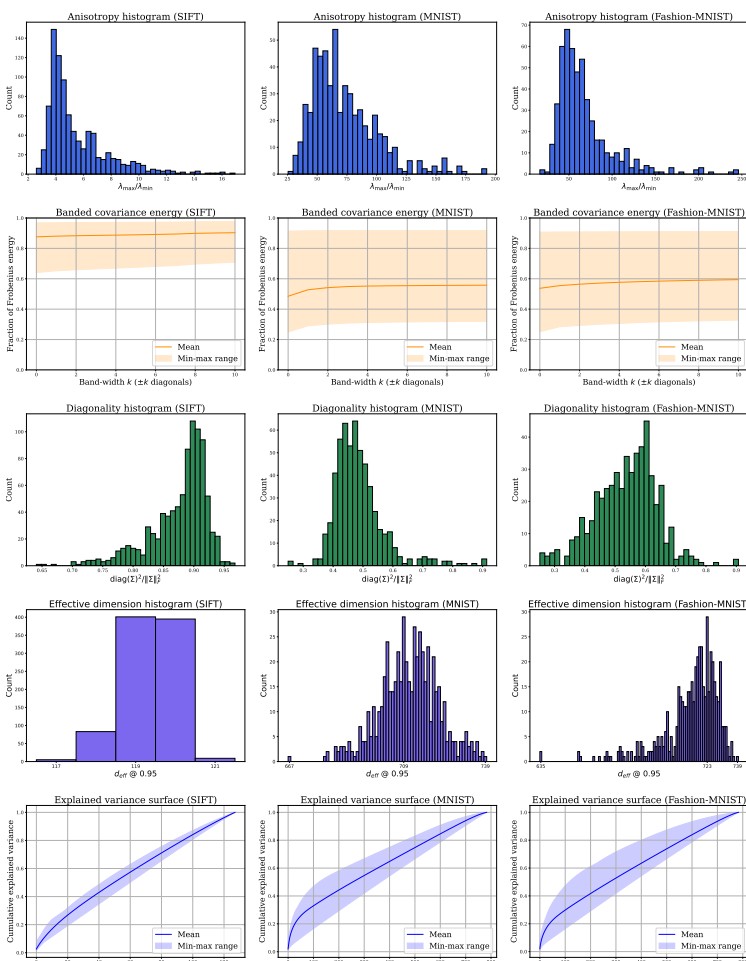

Figure 10: Spectral diagnostics across datasets (SIFT, MNIST, Fashion-MNIST; columns) and Gaussian properties (rows). Each subplot shows a different measure: anisotropy, band energy, diagonality, effective dimension, and explained variance.

(first row) reveal how Gaussians adapt to regions of different local dimensionality, with $\lambda_{\max}/\lambda_{\min}$ ratios ranging from nearly isotropic to highly stretched configurations across all datasets. Rather than simply partitioning space uniformly, Gaussians adapt their shapes to the underlying manifold structure, as confirmed by the diagonality histograms (third row) and effective dimension measurements (fourth row). This representation allows us to model stratified data where different intrinsic dimensionalities coexist, allowing the data to become pancake-like for surface regions, needle-like for curve regions, and ball-like for volumetric areas. Unlike traditional space partitioning methods, GARLIC models the underlying data distribution probabilistically, not just approximating distances for retrieval. The explained variance surfaces (bottom row) show that Gaussians efficiently capture the relevant dimensions at each location, enabling estimation of true manifold distances rather than just Euclidean distances to samples. This provides more semantically meaningful results in regions where the intrinsic dimensionality is lower than the ambient space. This completes the supplementary material.

## A.2 IMPLEMENTATION DETAILS

This subsection summarizes all implementation and training-specific parameters used in our model, including optimizer schedules, architectural constants, and adaptive procedures such as splitting, cloning and pruning. These details provide context for reproducibility and

support the complexity analysis in the main paper. Furthermore, there are dataset specifics, such as licenses and descriptions.

Table 3: Dataset Information

| Dataset | Dimension | Size | Description | License |
|---------|-----------|------|-------------|---------|
| SIFT1M | 128 | 1M | Image descriptors | CC0 |
| MNIST | 784 | 70K | Handwritten digits | CC BY-SA 3.0 |
| Fashion-MNIST | 784 | 70K | Fashion items | MIT |

**Datasets.** We evaluate our method on three standard benchmark datasets: (1) SIFT1M Lowe (2004), containing one million 128-dimensional SIFT descriptors that capture scale- and rotation- invariant local image features; (2) MNIST LeCun et al. (1998), consisting of 70,000 grayscale handwritten digit images (28×28 pixels) flattened to 784-dimensional vectors; and (3) Fashion-MNIST Xiao et al. (2017), a more challenging variant with the same format but featuring 10 categories of fashion items. For retrieval tasks, we used the ANN-Benchmark Aumüller et al. (2020) versions of these datasets (available at `http://ann-benchmarks.com/sift-128-euclidean.hdf5`, `http://ann-benchmarks.com/mnist-784-euclidean.hdf5`, and `http://ann-benchmarks.com/fashion-mnist-784-euclidean.hdf5`) to ensure a standardized comparison with existing methods.

**Experimental setup.** Our method was implemented in Python, using optimized libraries such as `PyTorch` Paszke (2019) and `NumPy` Harris et al. (2020). The experiments were carried out on an Intel Core i7-7820X CPU (16 threads), a Quadro RTX 8000 GPU (48 GB VRAM) and 125 GiB of RAM.

**Training configuration.** Batch sizes of 50000 are used for SIFT, and 20000 for MNIST and Fashion-MNIST datasets, with z-score normalization by subtracting the mean and dividing by the standard deviation. The number of training epochs is 250, with a typical early stop at 120. Gaussian updates are scheduled with two phases: a warm-up phase lasting 35 epochs and an optimization phase where structural operations like splitting and cloning are triggered every 35 epochs and pruning every 60.

**Learning rate schedule.** The learning rates follow a linear warm-up and exponential decay scheme. Specifically, for the Cholesky parameters, the rates are subjected to a warm-up phase from $1 \times 10^{-7}$ to $5 \times 10^{-4}$, followed by a decay to $9 \times 10^{-5}$. In terms of the means, they are warmed up from $1 \times 10^{-7}$ to $9 \times 10^{-3}$, and subsequently decay to $3 \times 10^{-3}$. Notably, the learning rate associated with the means is maintained at a relatively higher level than that of the covariances. This approach is designed to promote the adjustment of the Gaussian centers rather than the expansion of their radii.

**Adaptive refinement.** Splitting is applied to Gaussians with cardinality exceeding a fraction of the dataset $\gamma = 1 \times 10^{-2}$, using DBSCAN or K-Means, as fallback, with $c = 2$ clusters. The covariance of each new Gaussian is scaled down by $9 \times 10^{-1}$. Cloning selects dense regions just outside the Gaussian boundary, defined by a Mahalanobis shell with inner threshold $\tau$ and outer threshold $(1 + e) \cdot \tau$, where $e = 2.2$ controls the shell thickness. From the set of points referenced in $(\tau, e\tau]$, a random selection of 60% is made. Cloning is not performed on a Gaussian unless its cardinality surpasses a threshold specified by $8 \times 10^{-4}|X|$. Gaussians that have degenerated into a single point are eliminated. Pruning is executed at intervals of every 60 epochs.

**Quantization.** Local PCA is performed per cell using top-3 eigenvectors. Reduced points are quantized using a spherical grid with $n_{\text{radial}} = 6$ and $n_{\text{angular}} = 4$, forming directional bins per Gaussian.

**Loss.** The total loss is a weighted sum of three components: divergence $\lambda_{div} = 1.0$, covariance $\lambda_{cov} = 1.0$, and anchor term $\lambda_{anchor} = 10^{-2}$, with a weight $\alpha = 10^{-1}$ that balances position and shape. When calculating the $\mathcal{L}_{cov}$ loss, a numerical epsilon of $1 \times 10^{-12}$ is used to ensure stability.

### A.3 Complexity Analysis

We analyze the computational time and space complexity of our method in three parts: index construction, query execution, and storage. The analysis is expressed in terms of standard parameters, including the dataset size $|X|$, embedding dimension $d$, the number of Gaussians $K$, and the reduced PCA dimension $r \ll d$. Our goal is to ensure that each component remains scalable with respect to high-dimensional data and large-scale datasets. We summarize the complexity of each phase below.

**Index build complexity.** Let $I$ be the number of optimization steps and $K'$ the initial number of Gaussians. We denote by $S$, $C$, and $P$ the number of splits, clones, and pruned Gaussians, respectively, and define the final number of Gaussians as $K = K' + S + C - P$. Let $|B_g|$ be the average cell size, $c$ the number of K-Means clusters used during splitting, and $k'$ the number of candidate points sampled per cloning operation.

For the initialization, since we use K-Means++ on $K'$ total cluster centers, we need $\mathcal{O}(K' \cdot d \cdot |X|)$ time. For the optimization part, we need to perform a total number of $I$ iterations of full Mahalanobis-based point-to-Gaussian assignment, thus a total of $\mathcal{O}(I \cdot |X| \cdot K \cdot d^2)$ worst-case time. Separate from the optimization, we analyze the split, clone and prune operations that are not applied on every iteration of the optimization. (i) The split operation runs DBSCAN or K-Means (with $c$ clusters), thus for a total of $S$ such operations we would need $\mathcal{O}(S \cdot (|B_g| \cdot d \cdot c + d^2))$; (ii) the clone operation locates the subset of points outside the Gaussian's boundary (between $\tau \cdot \sigma$ and $(1 + \epsilon) \cdot \tau \cdot \sigma$) for which it identifies new local modes, thus for a total of $C$ operations, it leads to $\mathcal{O}(C \cdot (k'^2 \cdot d + d^2))$; and (iii) the prune operation simply removes low-cardinality Gaussians and reassigns the points to the nearest active Gaussian, which takes $\mathcal{O}(P \cdot |B_g| \cdot d)$ time. For the quantization of each Gaussian, PCA is performed on all points inside the Gaussian, which projects the data into reduced local bases. In total, for the quantization we need $\mathcal{O}(K' \cdot |B_g| \cdot d^2 + |X| \cdot d \cdot r)$ time. From all the terms described, the optimization term dominates.

**Query complexity.** Let $K$ be the number of Gaussians, $k$ the number selected per query, $d$ the dimension, $r$ the PCA dimension, $b$ the number of bins per Gaussian, $T$ the number of optimization steps to find the shortest distance from the query point to the boundary of a spherical bin in the reduced PCA space, $\rho$ the probed bin ratio, and $\beta$ the average bin size. For a single query, we first need to measure distances from the set of Gaussians, which takes $\mathcal{O}(K \cdot d^2)$. Then, for the $k$ nearest Gaussians we need to locate the subset of data to be examined. For this, for each of the selected $k$ Gaussians, we need to compute the local PCA projections ($\mathcal{O}(k \cdot d \cdot r)$), then compute and sort the spherical distances to all $b$ bins ($\mathcal{O}(k \cdot b \cdot r \cdot T)$), of which only the $\rho$ fraction is probed. From each, up to $\beta$ candidates are gathered and re-ranked using Euclidean distance, which needs $\mathcal{O}(k \cdot \rho \cdot b \cdot \beta \cdot d)$ time. In practice, the re-ranking factor dominates the complexity, which is sublinear.

**Space complexity.** Let $K$ be the number of Gaussians, $d$ the data dimension, and $N = |X|$ the dataset size. The model stores mean vectors $\boldsymbol{\mu} \in \mathbb{R}^{K \times d}$, Cholesky parameters $\mathbf{L} \in \mathbb{R}^{K \times d \times d}$, and cells storing point indices, requiring $\mathcal{O}(N)$ space. Thus, the total space complexity is:

$$\mathcal{O}\left(K \cdot (d^2 + d) + N\right)$$

where $K \cdot d^2$ dominates. Still, space complexity can be reduced to $\mathcal{O}(K \cdot d)$ by enforcing diagonal covariance matrices, at the expense of reduced expressiveness in anisotropic regions of the space.

