# OpenReview forum: "GARLIC: GAussian Representation LearnIng for spaCe partitioning"
_ICLR.cc/2026/Conference — ICLR 2026 Conference Withdrawn Submission_

### Official Review · Reviewer_rZt7 · 2025-10-17

**Soundness:** 2
**Presentation:** 2
**Contribution:** 2
**Rating:** 2
**Confidence:** 5

**Summary:**

The task that is solved is partition-based approximate nearest neighbor (ANN) search, where a space partitioning, such as $k$-means clustering is used as an index structure. In the query phase, the query point is routed into the closest clusters, and the nearest neighbors (usually after further pruning, for instance, via product quantization) of the query point from these clusters are returned as the approximate nearest neighbors.

The article proposes a novel partitioning method for this task. In particular, a partitioning based on a Gaussian mixture model is proposed for clustering the data points. This model is learned by minimizing an information-theoretic objective that balances coverage, assignment confidence, and structural consistency. Further, the clusters can be adapted to the data distribution by _splitting_ and _cloning_ operations. At each cluster, the data points are quantized by first projecting them onto the local principal component directions, and then binning along their hyperspherical coordinates.

In the query phase, the closeness of the clusters to the query point is defined by the Mahalanobis distance. The points of the selected clusters are further pruned by the aforementioned quantization scheme. Finally, exact distances from the query point to the pruned cluster points are evaluated and the nearest points are returned as the approximate nearest neighbors.

The experimental results of the article demonstrate that the proposed method is competitive with other space partitioning methods for ANN search, such as IVF-PQ and IVF-PQS, on MNIST, Fashion-MNIST, and SIFT data sets, when the efficiency is measured by the number of distance computations. The proposed method performs particularly well on the low-recall regime.

**Strengths:**

- [S1] The proposed partitioning scheme seems to adapt well to the underlying data distribution

- [S2] The explanation why the proposed partitioning scheme adapts to the underlying data distribution is intuitive.

**Weaknesses:**

- [W1] The authors claim that they follow the standard setup of the ANN-Benchmarks (Aumüller et al., 2020). However, ANN-benchmarks use queries-per-second (as measure by wall clock time) to measure the efficiency of the algorithms, whereas the authors use the number of distance computations as a proxy for efficiency. The authors justify this by claiming (lines 330-332) that the dominant inference cost comes from the distance computations of the reranking phase, and refer to the complexity analysis presented in Appendix A.3. However, the complexity analysis of Appendix A.3. does not justify this claim, but only states that "in practice, the reranking factor dominates the complexity" (lines 1067-1068). This is circular reasoning: the empirical analysis justifies the claim by referring the complexity analysis, and the complexity analysis justifies the claim by referring to the empirical results (N.B., the article does not contain an ablation experiment measuring the costs of the components of the inference). Since the GMM proposed in the article for space partitioning is more complicated than the baseline partitioning methods, such as $k$-means, it is not a fair comparison when the efficiency is reduced to the number of distance computations. In particular, computing the distances to the cluster centroids in $k$-means is only a $\mathcal{O}(Kd)$ operation, whereas in the proposed method it is a $\mathcal{O}(Kd^2)$ operation (see Appendix A.3., lines 1062 - 1063). In summary, wall-clock time is the gold standard for measuring the efficiency of ANN algorithms (see, e.g., the following recent benchmarks: Aumüller et al., 2020, Jääsaari et al., 2025, Kang et al., 2025), and using the number of distance computations as a proxy hides costly inference steps.

- [W2] The proposed method contains two components: a novel space partitioning scheme (that corresponds to the IVF part of the IVF-PQ) and a novel quantization scheme (that corresponds to the PQ part of the IVF-PQ). The main claim of the article is the effectiveness of the proposed space partitioning scheme for ANN search. However, the experiments of the article contains only an end-to-end evaluation (with a faulty proxy metric, see [W1]) of the method that contains both the partitioning method and the quantization method. More fine-grained experiments that compare (a) different space partitioning methods while keeping the quantization method constant (or simply by skipping the quantization step by directly reranking the points of the selected clusters); and b) different quantization methods for further pruning of cluster points by keeping the partitioning method constant (e.g. by combining all the quantization methods with $k$-means) should be performed to find out whether a) the proposed partitioning method is more efficient than the baseline partitioning methods; and b) the proposed quantization method is more efficient than the earlier quantization methods. The efficiency should be measured by queries-per-second (measured by wall-clock time, see [W1] for a further discussion).

- [W3] The experiments are done using only three data sets (MNIST, Fashion-MNIST, and SIFT) that have outdated representations: no one performs ANN search on raw pixels or SIFT descriptors anymore. See, e.g., Jääsaari et al., (2025), Kang et al., (2025), or Simhadri et al., (2024) for embedding data sets that are used in the modern applications of ANN search, such as retrieval-augmented generation.


References:

Aumüller, Martin, Erik Bernhardsson, and Alexander Faithfull. "ANN-Benchmarks: A benchmarking tool for approximate nearest neighbor algorithms." Information Systems 87 (2020): 101374.

Jääsaari, Elias, et al. "VIBE: Vector Index Benchmark for Embeddings." arXiv preprint arXiv:2505.17810 (2025).

Kang, Guoxin, et al. "BigVectorBench: Heterogeneous Data Embedding and Compound Queries are Essential in Evaluating Vector Databases." Proceedings of the VLDB Endowment 18.5 (2025): 1536-1550.

Simhadri, Harsha Vardhan, et al. "Results of the Big ANN: NeurIPS'23 competition." arXiv preprint arXiv:2409.17424 (2024).

**Questions:**

I do not have any questions concerning the manuscript.

---

### Official Review · Reviewer_2r6q · 2025-10-23

**Soundness:** 1
**Presentation:** 2
**Contribution:** 2
**Rating:** 2
**Confidence:** 5

**Summary:**

This paper introduces GARLIC, a representation learning framework for Euclidean approximate nearest neighbor (ANN) search that partitions the space into anisotropic Gaussian cells. Unlike isotropic or fixed-resolution methods (e.g., IVF, k-Means), GARLIC learns Gaussian components whose covariances adapt to local data density and geometry. The model is trained using a three-term loss — divergence, covariance, and anchor losses — to balance coverage, confidence, and geometric alignment.
At query time, the algorithm selects Gaussians via Mahalanobis scores, projects queries into local PCA subspaces, and retrieves candidates from locally quantized bins. Experiments on SIFT1M, MNIST, and Fashion-MNIST show competitive recall – #distances trade-offs.

**Strengths:**

- The paper is clearly motivated, addressing the issue of isotropic partitions in traditional ANN methods.
- The framework combines Gaussian parameterization, information-theoretic objectives, and adaptive refinement (split/clone), which are conceptually interesting.
- Experimental comparisons include a variety of ANN baselines (k-Means, LSH, PCA Tree, Faiss-IVF, IVFPQFS), showing reasonable empirical coverage.

**Weaknesses:**

**W1. Unclear interaction between loss components and retrieval performance**

The total loss L = L_div + L_cov + L_anchor combines three objectives, yet the paper does not provide an analytical or empirical explanation of how these terms interact with retrieval quality.
From Table 1, L_div or L_cov can improve coverage, but it is unclear whether improving these terms jointly leads to better query performance or conflicts with L_anchor, as the mutual influence among the three losses remains underexplained. This makes the theoretical justification for the composite objective incomplete.

**W2. Too many hyperparameters without guidance or sensitivity analysis**

The method introduces a large number of hyperparameters (approximately 16) spanning loss weights (e.g. λ_{div, cov, anchor}, α, τ), split/clone thresholds (γ, ρ, e, k) in refinement controls, and others in quantization and inference.
There is no clear guideline for how to tune or prioritize them. While some ablations (Fig. 4e–f, Fig. 7c) explore a few parameters, others (e.g., λ, α, ρ, β) are fixed without justification. The paper lacks discussion on which parameters dominate performance or affect convergence, leaving the method difficult to reproduce and tune for new datasets.

**W3. Unrealistic performance metric and cost accounting**

The efficiency is reported solely in terms of the number of distance computations, while major query-time costs are ignored. Querying involves multiple heavy operations — computing Mahalanobis distances for all Gaussians, projecting queries into each Gaussian’s local PCA subspace, and converting coordinates into hyperspherical bins.
These steps dominate runtime but are not captured by the “distance count” metric. As such, the claimed efficiency gains may not translate into real latency improvements. The reliance on a hardware-independent metric is conceptually clean but practically misleading given these expensive preprocessing steps.

**W4. Weaker performance than simple baselines at high recall**

In Fig. 3, k-Means and Faiss-IVF outperform GARLIC when Recall > 90% under the same number of distance computations. This suggests that while GARLIC is efficient at low recall, it loses its advantage when higher accuracy is required. The method’s complexity (Mahalanobis evaluation, projection, quantization) does not translate into a consistent benefit across regimes. Thus, the improvement seems marginal and does not convincingly surpass simpler, well-understood baselines.

**Questions:**

Please address the raised weakness points above, and further questions below

**Q1. Query-time cost breakdown.**

Could the authors provide a detailed decomposition of the query runtime into its main components:

(1) computing Mahalanobis scores to identify the nearest Gaussians,

(2) projecting the query into local PCA subspaces,

(3) computing distances between the projected query and bins, and

(4) evaluating the final exact distances?

This breakdown would clarify which step dominates runtime and whether the “#distance computations” metric reflects total query cost.

**Q2. Cloning and splitting conditions.**

What are the exact conditions that trigger Gaussian cloning and splitting, and how are thresholds (e.g., covariance factor
α) determined? If a split cannot sufficiently reduce the covariance to the specified factor, what fallback or termination criterion is applied? Please clarify how these operations affect training stability and partition granularity.

**Q3. Motivation for partial-data baselines.**

In Fig. 4a, Faiss-IVF appears to run significantly faster than GARLIC. What is the motivation for evaluating Faiss-IVF on 1% or 5% subsets of the dataset, given that it already scales efficiently on the full set?
A clarification of whether this subsampling serves fairness in runtime comparison or compensates for implementation constraints would be helpful.

---

### Official Review · Reviewer_cXiN · 2025-10-29

**Soundness:** 1
**Presentation:** 3
**Contribution:** 1
**Rating:** 2
**Confidence:** 3

**Summary:**

The paper proposes a new indexing scheme based on anisotropic partitioning of the search space into several cells and only a subset of cells are traversed during the search stage.

**Strengths:**

I believe the main motivation of the proposed indexing scheme (the need of anisotropic cells) is valuable  and can potentially lead to new approaches in the future research.

**Weaknesses:**

(1) All the experiments are performed on outdated benchmarks. Real-world applications do not use SIFT/flattenedMNIST in 2025, therefore the results are not informative for the practitioners, at least Deep1M should be used.

(2) The authors do not report the comparison to the alternatives in terms of wall-clock time, only in terms of accuracy-vs-number_of_retrieved_candidates. But the complexity of retrieving the candidates can be very different for various methods. For instance, I assume that computing the Mahalanobis distances in the authors' method can be expensive. Without the comparison in terms of wall-clock time there would be no sufficient evidence to justify the applicability of the method.

(3) I did not understand why the authors did not compare with graph-based methods. Yes, they induce certain space overheads but the authors' method is also not lightweight (e.g. it has to store a large number of covariance matrices). The authors should compare their method to the HNSW under exactly the same memory budget.

**Questions:**

Please, address my concerns in the Weaknesses section.

---

### Official Review · Reviewer_jedp · 2025-10-31

**Soundness:** 1
**Presentation:** 2
**Contribution:** 1
**Rating:** 2
**Confidence:** 4

**Summary:**

The authors propose a new partition-based method for approximate nearest neighbor search, where partioning is done using Gaussian cells. This partioning is described as being more geometry-aware than a regular $k$-means clustering of the data since the individual Gaussians can adapt to the local densities.

The proposed method is learned through optimizing an objective function that balances three objectives (reconstruction and two different regularization objectives). In the experiments, the authors compare the recall versus the number of distance computations for a number of partition-based baseline methods. The results show that, at lower recall levels, the proposed method can achieve the same recall using less distance comparisons than regular $k$-means based partitioning.

**Strengths:**

The authors aim to improve the partitioning itself in partition-based ANN methods which has recently been an understudied subject due to the focus shifting to graph-based ANN methods, but is still an important research area. The authors propose a several promising tweaks to make the method work, and the effects of these tweaks are studied in ablation studies.

**Weaknesses:**

The method proposed by the authors is a rather convoluted method for which the authors' own experiments do not provide indication that the proposed method is actually useful in practice. Comparisons are performed on three datasets, two of which (MNIST and Fashion-MNIST) contain only 60K points. It is very easy to achieve Recall@1 > 0.9 on these datasets, and in most practical scenarios this is the only regime of interest, yet the proposed method actually performs the same or worse than the simple $k$-means method in this regime for all datasets! In addition, it is not clear how the method performs for larger values of $k$.

The used baselines are weak, as the best performing baselines are basic IVF implementations. The experiments are missing comparisons against several state-of-the-art partition-based methods such as BLISS [1], ScaNN [2], and LoRANN [3] (although the latter two are orthogonal to the partitioning, but the authors also compare e.g. to Faiss-IVFPQFS). Also, while I agree that it is meaningful to compare the number of distance computations, it is also crucial to consider the end-to-end query time. The proposed query method involves selecting the closest Gaussians using Mahalanobis distance and computing PCA projections, which in practice will significantly slow dow the query process. For example, in the experiments Faiss-IVFPQFS performs worse than Faiss-IVF, while in practice on the exact same datasets Faiss-IVFPQFS should have better end-to-end performance [4]. While comparing end-to-end performance is dependent on hardware and implementation details, it would be easy to at least compare against an IVF implementation on equal footing. To demonstrate the true potential of the method, the authors should also aim to study a query implementation of their method that is as optimized as possible such that it could be fairly compared to methods in e.g. Faiss that can be assumed to also be as optimized as possible.

The authors do not report indexing times and scalability to datasets larger than 1 million points is not demonstrated. The assignment step (which is the dominant cost) in each optimization iteration requires $\mathcal{O}(NKd^2)$ time, which is $d$ times slower than the assignment step in $k$-means and the method requires more iterations than $k$-means (typically less than 20 with large $k$). The simplicity and scalability of $k$-means clustering to large datasets is its major benefit, and $k$-means is also applicable to inner product measures using spherical $k$-means. In addition, GARLIC features several hyperparameters that need to be tuned, and it is not clear how stable training the method for a new dataset is with respect to e.g. the hyperparameters balancing the different loss terms, the learning rate, the batch size, etc.

[1] Gupta et al. BLISS: A Billion Scale Index Using Iterative Re-Partitioning. KDD, 2022.

[2] Guo et al. Accelerating Large-Scale Inference with Anisotropic Vector Quantization. ICML, 2020.

[3] Jääsaari et al. LoRANN: Low-Rank Matrix Factorization for Approximate Nearest Neighbor Search. NeurIPS, 2024.

[4] Aumüller et al. ANN-benchmarks: A benchmarking tool for approximate nearest neighbor algorithms. Information Systems, 2020.

**Questions:**

- How do the authors hypothesize that the end-to-end performance of their method compares against e.g. IVF?

- How does the indexing time of your proposed method compare to IVF? Is training the model on large datasets feasible without a GPU?

---

### Note · Authors · 2025-11-14

**Comment:**

We thank the reviewers for their feedback, and we will incorporate it in a revised version.

**Withdrawal Confirmation:**

I have read and agree with the venue's withdrawal policy on behalf of myself and my co-authors.